# Mayflies (Ephemeroptera) and Their Contributions to Ecosystem Services

**DOI:** 10.3390/insects10060170

**Published:** 2019-06-14

**Authors:** Luke M. Jacobus, Craig R. Macadam, Michel Sartori

**Affiliations:** 1Division of Science, Indiana University Purdue University Columbus, 4601 Central Ave., Columbus, IN 47203, USA; 2Buglife—The Invertebrate Conservation Trust, Balallan House, 24 Allan Park, Stirling, Scotland FK8 2QG, UK; craig.macadam@buglife.org.uk; 3Musée cantonal de zoologie, Palais de Rumine, Place de la Riponne 6, CH-1005 Lausanne, Switzerland; michel.sartori@vd.ch; 4Department of Ecology and Evolution, University of Lausanne, Biophore, CH-1015 Lausanne, Switzerland; michel.sartori@unil.ch

**Keywords:** aquatic insects, diversity, adaptations, ecology, freshwater systems, terrestrial systems

## Abstract

This work is intended as a general and concise overview of Ephemeroptera biology, diversity, and services provided to humans and other parts of our global array of freshwater and terrestrial ecosystems. The Ephemeroptera, or mayflies, are a small but diverse order of amphinotic insects associated with liquid freshwater worldwide. They are nearly cosmopolitan, except for Antarctica and some very remote islands. The existence of the subimago stage is unique among extant insects. Though the winged stages do not have functional mouthparts or digestive systems, the larval, or nymphal, stages have a variety of feeding approaches—including, but not limited to, collector-gatherers, filterers, scrapers, and active predators—with each supported by a diversity of morphological and behavioral adaptations. Mayflies provide direct and indirect services to humans and other parts of both freshwater and terrestrial ecosystems. In terms of cultural services, they have provided inspiration to musicians, poets, and other writers, as well as being the namesakes of various water- and aircraft. They are commemorated by festivals worldwide. Mayflies are especially important to fishing. Mayflies contribute to the provisioning services of ecosystems in that they are utilized as food by human cultures worldwide (having one of the highest protein contents of any edible insect), as laboratory organisms, and as a potential source of antitumor molecules. They provide regulatory services through their cleaning of freshwater. They provide many essential supporting services for ecosystems such as bioturbation, bioirrigation, decomposition, nutrition for many kinds of non-human animals, nutrient cycling and spiraling in freshwaters, nutrient cycling between aquatic and terrestrial systems, habitat for other organisms, and serving as indicators of ecosystem health. About 20% of mayfly species worldwide might have a threatened conservation status due to influences from pollution, invasive alien species, habitat loss and degradation, and climate change. Even mitigation of negative influences has benefits and tradeoffs, as, in several cases, sustainable energy production negatively impacts mayflies.

## 1. Introduction

Our contribution to the Diversity and Ecosystem Services special issue of this journal focuses on the amphinotic insect order Ephemeroptera and the varying—but specific—roles these diverse organisms play in providing direct and indirect services to humans and other parts of our global array of freshwater and terrestrial ecosystems. This work is intended as a concise overview that provides general examples, illustrations, and context. It is not exhaustive. We attempted to balance communicating the breadth of the extensive information available with providing a concise and readable summary.

Few other groups of insects may have such a variety of common names as Ephemeroptera. Though Ephemeroptera are generally referred to as mayflies, they sometimes are called dayflies, shadflies, or even fishflies. The names of particular species can be vividly descriptive [1], such as the Yellow may dun (*Heptagenia sulphurea* (Müller) (Heptageniidae)), or March brown (*Rhithrogena germanica* Eaton (Heptageniidae) and *Maccaffertium vicarium* (Walker) (Heptageniidae)), or of more obscure origin such as Jenny spinner (*Paraleptophlebia mollis* (Eaton) (Leptophlebiidae)) or Drake mackerel (*Ephemera vulgata* Linnaeus (Ephemeridae)).

As scientific pursuits grow in complexity and necessarily narrow in focus, we risk the danger of losing sight of the interconnections and broader implications of individual components of ecosystems. Mayflies spend the majority of their lives under water, out of human sight and mind. Thus, they are an ecosystem component that easily eludes the continued attention of those not a part of the “guilds” (borrowing a term from Niko Tinbergen’s Foreword to Dethier’s *To Know a Fly* [2], and later used here in another context) of aquatic biologists and others who spend time with them daily.

Mayflies are known best in popular culture for their short-lived adults, which may survive for only minutes [3]. They also are well-known for their mass emergences, which can cover areas tens or hundreds of square kilometers in size with swarms estimated to be 125–250 m thick; these are significant enough to be detected and tracked en masse on weather radar [4,5]. To the angler and ecologist [6] they are known for serving as patterns for imitations [7] and for being indicators of environmental quality [8], respectively. However, mayflies also warrant special attention for the many other important services they provide as constituents of freshwater and terrestrial ecosystems.

## 2. Biological Attributes of Mayflies

The provision of ecosystem services by mayflies is both enabled and constrained by the biological attributes of the organisms themselves.

### 2.1. Life Cycle and General Biology

Mayflies are unique among extant insects by having a subimago stage in their metamorphic cycle (Figure 1a). The subimago is an active and mobile stage that occurs between the ultimate larval instar and the mature adult stage, or imago, when present in the life cycle [9]; however, in a few rare cases, the subimago is the terminal stage in females (e.g., *Dolania* Edmunds & Traver (Behningiidae) [10], *Prosopistoma* Latreille (Prosopistomatidae) [11,12], or even in both sexes (e.g., some Palingeniidae [13]). This is a remarkable example of reproductive diversity, in that different stages of the order may be the mature stage of a particular species. In general, the subimago represents the transition from the aquatic to terrestrial parts of the mayfly species’ lifecycle, sometimes with very large numbers of individuals emerging simultaneously from the water (e.g., [14]). The subimago molts to the imago stage while resting on riparian vegetation or, rarely, in the air, mid-flight [15]. Subimagoes and imagoes (Figure 1b) are notoriously short-lived and therefore have a narrow window of time during which they provide ecosystem services. With non-functional mouthparts and digestive systems, the imago stage (and also to a certain extent the subimago) is specialized for dispersal and reproduction. Alate individuals usually have two pairs of wings, with the hind wings being smaller than the forewings (Figure 1); a few taxa have only the forewings. Alate stages (Figure 1) have two or three terminal filaments of varying relative lengths, depending on the taxon in question.

Mayfly species engage in both sexual and parthenogenetic (asexual) reproductive strategies [16]. Most species lay eggs under the water’s surface, though a very few are ovoviviparous (e.g., *Callibaetis* Eaton (Baetidae) [17], *Cloeon* Leach (Baetidae) [18]). Most species are probably univoltine in temperate regions, though many may have multiple generations per year, and a few may require two or three years to complete their life cycle.

The strictly aquatic larvae, also called nymphs, are comprised of several instars. The number of instars that occur during an individual’s life depends primarily on food resources [19] and water temperature (e.g., [20]). The larvae are common constituents of most freshwater biotopes, both lotic and lentic, and in some cases can tolerate brackish water at least temporarily [21,22]. The larvae are found on or in almost all submerged substrates. Their near ubiquity in freshwaters indicates how widely mayflies are contributing to ecosystem services. Their absence can be an indicator of problems in particular environments (see further discussion below). Fortunately, mayfly larvae are easily distinguished from other aquatic insects. The larvae of mayflies usually have three terminal filaments (e.g., Figure 2, Figure 3 and Figure 4), though some have two. Larvae have prominent forewing pads (Figure 3); hindwing pads are much smaller, sometimes being vestigial, or absent. Larvae have ten abdominal segments, with pairs of articulated gills on at most segments one through seven (Figure 3). The body lengths (excluding terminal filaments) of the vast majority of species fall between about 2 to 30 mm.

The larvae of mayflies can be assigned to a variety of feeding groups, or guilds, in aquatic systems. These feeding groups usually correspond to particular ecosystem services outlined elsewhere in this paper. A variety of morphological adaptations enable various behaviors and the penetration of specific microhabitats where services are carried out. Most taxa are collector-gatherers and scrapers, usually feeding on detritus and periphyton, with bacterial ingestion perhaps also playing an important role in their nutrition (e.g., [23]). These functions are facilitated by specialized mouthparts. Scrapers, for example, often have bladelike mandibles. In swift-water habitats, a variety of abdominal adaptations, such as friction disks formed by setae (e.g., some *Drunella* Needham (Ephemerellidae)*,*
Figure 2) and gills (e.g., *Rhithrogena* Eaton (Heptageniidae), Figure 4), enable individuals to maintain their purchase while feeding [24]. These collector-gatherer and scraper taxa are generally clinging, sprawling, and swimming in habit, with these tendencies facilitated by the morphology of their claws and orientation of legs, generally flattened bodies (Figure 3), or streamlined bodies, respectively.

Several taxa, such as species of the genera *Coloburiscus* Eaton (Coloburiscidae), *Isonychia* Eaton (Isonychiidae), and *Oligoneuria* Pictet (Oligoneuriidae) (Figure 5) are filter feeders and may use rows of setae on their forelegs to capture particulate matter passively; others, such as *Arthroplea* Bengtsson (Heptageniidae) may use specialized maxillary palps (Figure 6) to create vortices that actively concentrate particles from the water column [25], while others yet, such as *Ametropus* Albarda (Ametropodidae), may use the forelegs to create such a vortex [26]. Burrowing mayflies have legs and tusks modified for digging (Figure 7 and Figure 8) in a variety of benthic habitats including clay banks and heavy muck, and they use their gills to help move water through their burrows in order to collect food items [27]. Very few of these burrowing taxa, such as *Povilla adusta* Navás (Polymitarcyidae), bore into rotting wood, sometimes causing problems with human structures submerged in water [28]. Other mayfly taxa live and forage on or among sediments; these taxa often have operculate gills (Figure 9) that protect subjacent gills (e.g., most Caenidae and some Ephemerellidae) or modified abdominal segments that protect the gills (e.g., *Hyrtanella* Allen & Edmunds (Ephemerellidae), *Machadorythus* Demoulin (Machadorythidae) Figure 10).

Other taxa are predators (Figure 11), usually feeding on Chironomidae (Diptera) larvae or other mayflies (e.g., [29]). A few taxa may be leaf and vegetation shredders [30] (Figure 12). Rarely, mayflies are considered parasitic, such as the case of *Symbiocloeon* Müller–Liebenau (Baetidae) on mussels. However, this relationship actually is commensal in nature [31]. The association of mayflies with mussels in this way is a remarkable parallel coincidence when one considers the early developmental and dispersal biology of mussels, which involves attachment of larvae (glochidia) to fish gills.

In many cases, the feeding preferences and general habits of mayfly taxa are assumed based on their morphologies [30] and have not been observed directly. Expanding our knowledge of the microhabitats, movement, and feeding behaviors of mayflies will help us also to expand our knowledge of the precise ecosystem services provided by mayflies.

### 2.2. Global Diversity and Distribution

Ephemeroptera constitutes a small order of amphinotic insects associated with freshwaters worldwide, having fewer than 3700 species currently recognized. With origins more than 300 million years ago [32], it is one of the oldest groups of extant insects and has a relatively long history of contributing to planetary function.

The monophyly of Ephemeroptera is well-established, but its relationship with Odonata and Neoptera is not yet consistently resolved (see e.g., [33,34,35]). The phylogenetic relationships among mayflies themselves also remain unclear and partially unresolved. Resolution of deep levels of mayfly phylogeny remains elusive, confounding efforts to form consistent hypotheses about the groups’ adaptive radiations and the general evolution of ecological roles and associated ecosystem services. Based on molecular data, however, the monotypic family Siphluriscidae is hypothesized to be basal and the sister-group to all others [36], which is consistent with its many plesiomorphic features. Some families cluster together with confidence, such as those included in Ephemerelloidea, Caenoidea, and Fossoriae. Relationships within these broad groupings are not well understood at the moment, however. Additional molecular, morphological, and behavioral data may shed light on some of these longstanding problems.

Mayflies can be found in freshwater ecosystems distributed throughout the world. As stated previously, their near ubiquity is evidence of their wide contribution of ecosystem services, even though they are a relatively small order. They are absent only from Antarctica and some remote ocean islands, such as the Tristan da Cunha archipelago [37] or the Falkland/Malvinas Islands [38]. Some arctic and subarctic islands have very low diversity, such as Greenland and Iceland, which are inhabited by a single species each [39,40].

It has been generally hypothesized that mayflies are poor dispersers, due to their low vagility and short alate lifespans previously discussed. Recent studies, however, found that some oceanic and volcanic islands, such as Macaronesia [41,42] and the West Indies [43], have been colonized with subsequent in situ radiation. Continental islands, such as those that broke off of Gondwana, have unique lineages; examples of this kind of island include Madagascar, New Caledonia, and Seychelles.

About 20 species generally present Holarctic distributions, either Circumarctic (e.g., *Nixe joernensis* (Bengtsson) (Heptageniidae), the Eurasian flat-headed mayfly) or Transpacific (e.g., *Baetis bicaudatus* Dodds (Baetidae), Dodds’ small minnow mayfly), whereas more than 40 species have a Pan-American distribution as a result of the Great American Interchange [44]. Virtually all species from the Oriental, Afrotropical, Australasian, and Pacific Realms are endemic. The highest generic endemism is found in the Pacific, Afrotropical, and Australasian Realms.

The highest mayfly diversity is found in the Neotropics with almost 900 described species, followed by the Palaearctic (830), Nearctic and Oriental (610 and 620 respectively), Afrotropical (440) Australasian (250), and Pacific (48) (data updated from [45]). In general, diversity is greatest in the intertropical region and decreases towards the poles. Diversity also generally decreases with an increase in altitude.

Our knowledge about the diversity of mayflies worldwide is still increasing year after year. Since the last assessment [45] (about four years ago at the time of this writing), more than 300 new species have been described, and many still await description, especially in the tropics.

The composition of Ephemeroptera families and genera have changed greatly during the last 25 years, mainly because of the gathering of species into a more phylogenetic system. These narrower concepts of genera and families better reflect the diversity of the order in terms of how form, function, and ecosystem services align. Currently, mayfly species are spread among about 40 families (depending on the classification system used) and more than 460 genera. A few families are monogeneric and monospecific, namely Austremerellidae (Australia), Machadorythidae (West Africa), Rallidentidae and Siphlaenigmatidae (New Zealand), and Siphluriscidae (China). More than half of the families contain fewer than 20 species. The three richest families—Baetidae (110 genera, 1070 species), Heptageniidae (37, 606), and Leptophlebiidae (147, 718)—encompass two-thirds of all known species. On the generic level, four genera possess more than 100 species each: *Caenis* Stephens (Caenidae) with 164 species, *Baetis* Leach (Baetidae) with 158 species, *Rhithrogena* (Heptageniidae) with 155 species, and *Labiobaetis* Novikova & Kluge (Baetidae) with 104 species. No genus has a true cosmopolitan distribution. Among the most widespread genera, *Caenis* and *Baetis* are mentioned from the Australasian Realm by single and dubious records, whereas *Choroterpes* Eaton (Leptophlebiidae) reaches only Sulawesi in Australasia [46]. *Labiobaetis* is only absent from the Neotropics, as was *Cloeon* until the otherwise Afrotropical species *C. smaeleni* Lestage (Baetidae) was recently discovered as introduced to Brazil [47].

Some families resemble one another very closely in different geographic regions, due to convergent evolution in different radiations of diversity. For example, the widespread family Leptophlebiidae contains species with forms of gills usually associated with other extant families [3,48].

## 3. Roles in Ecosystem Services

Ecosystem services are defined as the benefits that people obtain from ecosystems [49]. These services can be further categorized as either cultural, provisioning, regulating, or supporting services.

### 3.1. Cultural Services

Cultural services are non-material benefits and are typically difficult to define but include aesthetic, spiritual or religious values, inspirational and educational uses, and sense of place and cultural heritage [50].

The cultural significance of mayflies is important but easily overlooked, especially by Western cultures. Mayflies have been used to illustrate the fleeting and fragile nature of life in literature across cultures and throughout the ages. The earliest reference to mayflies in a written text can be found in the *Epic of Gilgamesh*, which dates from the 18th Century BC [51]. Mayflies and their short adult stage piqued the interest of early scientific writers such as Aristotle and Pliny the Elder. Chinese poets, such as Shi Su and An Liu, featured them in verse (see English translations included in [52]). Their importance in fly-fishing (see below) has led to their inclusion in a number of poems, including ‘To an old friend’ [53] which describes an angler and a trout waiting for the emergence of the first March brown mayfly of the year. Several music groups have had mayfly as part of their names.

The Tisza mayfly *Palingenia longicauda* (Olivier) (Palingeniidae) is celebrated in Hungary with an annual festival. In addition, a monument to the mayfly can be found at Szeged, and the Tiszavirág (i.e., blossom of the Tisza) bridge over the River Tisza at Szolnok is designed to resemble the mayfly in flight. Incidentally, the Tisza mayfly features in a Hungarian folk song as a symbol of a man’s love [54]. One of us (LMJ) has visited the annual Bay-Rama Fishfly Festival held in June in New Baltimore, Michigan, United States of America (USA) (Figure 13); it is an event originally organized more in spite of the mass mayfly emergences of Ephemeridae from Lake St. Clair, rather than celebrating them. Another annual festival focused on mayflies occurs annually in Alagoas state, Brazil. These are a few examples that demonstrate the global cultural awareness of mayflies and the rites of intensification associated with them.

Mayflies are prominent features in some marketing campaigns. In England, for example, public houses are named after mayflies, and a number of real ales are named after mayflies. Major corporations have also employed mayflies as a marketing tool. Nike released a range of ultralight running shoes called ‘Mayfly’ in 2003, while Vodafone ran an award-winning £100 million marketing campaign in 2006 encouraging viewers to be like a mayfly and ‘make the most of now’.

Mayflies have given their name to a number of vehicles. One of the first attempts at powered flight was made in 1908 in an aircraft called the Seddon Mayfly, and it was followed in 1910 by the Bland Mayfly, the first aircraft to be designed by a woman, Lillian Bland. Mayfly was used as the name for three British Royal Navy ships: a 1907 torpedo boat, an airship in 1911, and a Fly-class river gunboat in 1915.

The imitation of mayflies to catch fish dates back to the 1st century AD [55]. In 15th century Britain, patterns for artificial flies to mimic *Rhithrogena germanica*, *Electrogena lateralis* (Curtis) (Heptageniidae), and *Heptagenia sulphurea* were being published [56]. Fly-fishing is now a world-wide pastime enjoyed by millions of people. Various assessments have been made of the economic importance of fly-fishing. In the United Kingdom, for example, game fishing is worth £498 million to the economy annually [57,58]. In another example from the USA, some 25.4 million people are believed to participate in freshwater fishing, contributing US$31.4 billion to the USA economy [59].

### 3.2. Provisioning Services

Provisioning services describe material benefits obtained from ecosystems such as food for humans, safe freshwater, and genetic resources [50]. The following paragraphs describe the provisioning services in freshwater ecosystems provided by mayflies.

Despite their generally small size and delicate nature, mayflies are naturally high in protein, minerals, B vitamins, and essential amino acids, and low in fat [60] which makes them an important component of the human diet in some cultures. Human consumption of mayflies has been documented from 10 countries [61]. The mayflies of Lake Victoria are particularly important for local inhabitants. Swarms of *Caenis kungu* Eaton (Caenidae) and *Povilla adusta* are collected from along the shore-line, then either sun-dried and ground into flour, or made into a paste, and subsequently cakes and bread [60,62,63,64]. In a marketplace in a village on the East coast of Madagascar, one of us (MS) has seen baskets of *Elassoneuria* Eaton (Oligoneuriidae) larvae sold as “Mangoro River shrimps”. Swarms of *Plethogenesia* Ulmer (Palingeniidae) are also collected in Papua New Guinea before being cooked and eaten [54]. In Indonesia, the Muyu people collect spent mayflies from the surface of rivers and creeks using mosquito nets. The catch is packed in wild banana leaves and roasted on embers or heated in a pan before being eaten [65]. Whilst adult mayflies are more commonly collected and eaten, 17th century Incas are reported to have eaten larvae of *Euthyplocia* Eaton (Euthyplociidae) either raw or by incorporating them into a spicy sauce [66]. The Nyishi and Galo tribes of India also used roasted or boiled larvae of the genus *Ephemera* Linnaeus (Ephemeridae) to treat stomach disturbances [67]. Alate stages and larvae of *Teloganopsis jinghongensis* (Xu, You & Hsu) (Ephemerellidae) are eaten in China [68], having one of the highest protein contents by dry weight of any edible insect [69].

Mayflies are used for more than just food. Low molecular weight chitosan, which has antitumor activity [70], can be produced from their bodies. Living mayflies, such as the Triangle small minnow mayfly, *Neocloeon triangulifer* (McDunnough) (Baetidae), also have important uses. This species has become an important laboratory model organism, useful for advancing scientific endeavors and expanding human knowledge (e.g., see review in [71]).

### 3.3. Regulatory Services

Regulatory services are benefits obtained from processes such as the regulation of climate, water purification, and pollination [50]. Mayflies contribute, at least in small ways, to the regulatory services provided by ecosystems in that they process, break down and sometimes remove substances from water as larvae, as discussed elsewhere in this review. A number of mayfly genera, including some mentioned previously, filter fine particulate organic material from the water column as a source of food [72,73]. Mayflies remove substances from water when they emerge as subimagoes. Although many individuals return to the water and die, some are retained in terrestrial systems via predation or incidental death.

### 3.4. Supporting Services

Supporting services are necessary for the production of all other services and include nutrient cycling and primary production [50]. Mayflies provide many essential services that maintain and enhance ecosystem function. Burrowing species (e.g., Figure 8) such as *Ephemera danica* (Müller) (Ephemeridae), *Hexagenia limbata* (Serville) (Ephemeridae) (the Michigan Hex) or *Campsurus violaceus* Needham & Murphy (Polymitarcyidae) contribute to both bioturbation and bioirrigation by reworking sediments in rivers and flushing water through their burrows (e.g., [74]). In one study [75], *H. limbata* was found to be responsible for up to 98% of the volume of sediment disturbed in Lake Saint Joseph, Canada. Another study [76] found that bioirrigation and bioturbation by *Hexagenia* spp. in Lake Erie resulted in soluble reactive phosphorus flux in the water column.

As just indicated, mayflies contribute in various ways to nutrient cycling (and spiraling) and energy flow. In a few special cases, mayflies also make a contribution to the decomposition of coarse woody debris and vegetation. Species from the genera *Povilla* Navás (Polymitarcyidae) and *Asthenopus* Eaton (Polymitarcyidae) burrow into submerged and rotten wood and living plants such as *Typha* Linnaeus (Poales: Typhaceae) and *Eichornia* Kunth (Commelinales: Pontederiaceae) [77,78], sometimes also causing damage to underwater structures and boats. Filter feeding by mayflies [72] has been assumed to contribute to water purification.

The abundance of mayflies makes them an important part of the diet of many species other than humans, as detailed above. As many as 224 species—including a range of other invertebrates (especially species of Arachnida and Odonata), birds, lizards and other reptiles, amphibians, bats and other mammals—feed upon mayflies [79,80,81]. Arguably the most important of these predator-prey relationships is the contribution of mayflies to the diet of fish. Many people around the world rely upon freshwater fish as a source of subsistence, not only as food, but also as a driver of the local economy [82]. The contribution of mayflies to the diet of freshwater fish varies considerably by species. While some fish species casually or incidentally feed on mayflies, other fish species rely almost exclusively on mayflies. In one example, 98% of the diet of the Oscar cichlid (*Astronotus ocellatus* (Agassiz) (Chordata: Actinopterygii: Cichliformes: Cichlidae)) is made up of mayflies [83].

In being an integral part of the diet of fish and other aquatic animals, mayflies serve as a link in the flow of energy between primary producers and secondary consumers. They may, for example, scrape and ingest periphyton from submerged surfaces and then be eaten by fish or other predators. Mayflies also may collect fine particulate organic matter (FPOM) and make energy and nutrients contained therein available to higher trophic levels in the aquatic community.

Mayflies not only move nutrients within aquatic ecosystems, but they also move nutrients between them. This may be important for maintaining a variety of aquatic communities, especially if various climate forecasts hold true. Some migrations of mayflies may prove to deliver food subsidies from productive but warming river mainstems to cool but food-limited tributaries, enhancing the resilience of cool-water predators in warming river networks. *Ephemerella maculata* Traver (Ephemerellidae), for example, has been shown to engage in such movements, and it was more important than terrestrial invertebrate subsidies to the early growth of a trout species [84].

Mayflies also play an important role in the cycling and transfer of nutrients and carbon between aquatic and terrestrial habitats. Although mayflies grow and develop in aquatic habitats, terrestrial detritus is the dominant nutrient source for at least the abdomen, head, and wings of the burrowing mayfly species, *Ephemera danica* [85]. The sediment, periphyton, and seston also play roles in the intertwined sources of nutrients for the species. Winged mayflies emerge from the water and enter the terrestrial realm, where they may be consumed by many riparian species such as birds, bats, spiders, and lizards (e.g., [79]). This represents, at least in part, their role in transferring matter from aquatic to terrestrial systems. In this role, however, it is important to consider that they also serve as a “biotransporter” of potentially problematic substances, such as waterborne contaminants, to terrestrial ecosystems [86]. Black bears (*Ursus americanus*) have been observed feeding on piles of dead mayflies in Canada [87], but more study is needed on this and whether they play any significant role in transferring contaminants directly to large mammals.

Mayfly larvae also serve as habitat for other organisms. Mayfly species are known to host commensal bryozoans [88], protozoans [89,90], and chironomid midges [90,91,92,93,94]. They are also hosts to various fungi [95,96], nematodes [97,98,99,100,101,102,103,104], and trematodes [99,105].

Together with other macroinvertebrates, aquatic insects—especially mayflies—have become an important tool for monitoring the quality of freshwaters (e.g., see [106,107,108]) and their associated terrestrial riparian habitats, due to the essential roles these arthropods play in aquatic ecosystems, their sensitivity to change, and our increasing abilities to collect and identify them. Mayflies, at the order and various subordinate taxonomic levels, have become well-known for their use in these efforts (e.g., [109,110,111,112]), which are applied to the protection of both biodiversity and human water supplies throughout the world (e.g., [113,114,115,116,117,118,119,120,121,122]).

Mayflies fulfil the criteria for good indicators because they are: (i) abundant and sufficiently diverse in their habits and habitats, (ii) sensitive and predictable in their response to changes in environmental conditions, (iii) relatively easily sampled and identifiable to meaningful taxonomic resolutions, and (iv) bioaccumulate chemicals such that the pathways of toxins in the environment can be traced [123,124]. As biological indicators, their response to changing conditions is integrated over time and space, potentially lowering sampling effort and cost compared to the high intensity of sampling often required when relying on chemical variables to detect certain impacts [125]. An additional benefit, particularly in economically disadvantaged areas, is the lack of specialized equipment and supplies needed for collecting mayfly data, in contrast to chemical data [126].

As mentioned previously, mayflies are identified relatively easily to the order level. With minimal education, mayflies can be further identified to the family level, even in the field after being collected with simple nets or seines. A South African tool called “miniSASS” (www.minisass.org) is now implemented in some places on relatively remote Indian Ocean islands and allows a rapid measure of general river health and water quality by children or adults. In the UK, the Riverfly Partnership has developed a simple monitoring method, targeted at anglers and other river users, which is now being used by over 3000 volunteers to monitor around 800 sites [127]. As such, species of Ephemeroptera are arguably among the most useful bioindicators and provide a scoring index that is easy to use [128].

Mayfly community composition changes in response to alterations of a variety of environmental variables [129]. Their broad ranges of functional traits and differential tolerances to anthropogenic factors have been noted widely and are either used, or have the potential for use, in the development of biotic indices to monitor agricultural practices [130], organic pollution [108,131,132,133], eutrophication [134,135], flow quantity [136], acidification [137], mine drainage [138], drought [139], sediment and silt loading [129,140], pesticide pollution [141], physical habitat alteration [142], invasive species [120], blooms of cyanobacteria [143] and climate-change vulnerability, particularly in long-lived species, and those living at higher altitudes and other areas where periods of drought may increase [144,145,146]. Mayflies also respond to changes in temperature [147,148,149,150,151,152,153,154]. As ‘biosentinels’, aquatic insects have been used to monitor levels of a variety of heavy metals [155], methylmercury [156], mercury [157,158], selenite [159], and uranium [160] in freshwater ecosystems. In particular, larvae are sensitive to low levels of nitrates in the water [161,162,163] and also to changes in phosphorus [134]. In one example [164], phosphorus enrichment actually increased mayfly growth rates. Further work is required to determine whether the species’ sensitivities are due to the increased growth of bacteria on their bodies under these conditions [165] or to increased cyanobacterial populations, perhaps moreso than to the nutrients themselves. Regarding changes in pH, it has been found that differential sensitivities exist at the species level [166]. Acidification, in particular, may play a role in community structure [167].

## 4. Challenges to Mayfly Services

The continued provisioning of ecosystem services by mayflies is reliant on healthy populations of the species that deliver them. Global freshwater biodiversity is reported to have declined by 83% since 1970, with greatest losses in the Neotropics, Indo-Pacific, and Afrotropics [168]. Whilst this analysis did not include invertebrates, we see no reasons to suggest that similar declines have not occurred. Another study reported that general flying insect biomass has decreased by 76% or more in some areas [169]. Aquatic insect species, in particular, likely are experiencing similar declines. A survey of trout anglers in southern England reported a perceived 66% reduction in the number of freshwater insects emerging from chalk streams since the 1970s [170], and in a recent assessment of red lists based on the IUCN assessment criteria [171] from around the world, 15% of dragonflies and damselflies (Odonata) were found to be at threat of extinction [172]. Sound evaluations of the conservation status of mayfly species are generally lacking globally. A few areas, however, have conducted such evaluation. In Switzerland, for example, a recent red list evaluated 43% of Ephemeroptera as at least endangered [173], whereas in France, this proportion is 22% [174]. Differing criteria used for evaluation make direct comparisons between studies difficult, and methods should be evaluated carefully. Some apparently threatened species are found in habitats that are difficult to study, as is the case for South Carolina, USA [175], leading to the possibility that some species simply are under-represented in stream samples. Clearly, more work is needed towards assessing the conservation status of mayfly species worldwide, but the percentage of imperiled mayfly species may be very near to the percentage of threatened odonates mentioned above (15%). Besides the loss of species diversity, concern also should be directed towards decreases in genetic diversity, which may lead to populations that are less able to adapt to changing conditions and more prone to effects of genetic drift.

The following pressures should be managed to help conserve mayfly populations and by extension, the ecosystem services they provide.

### 4.1. Pollution

Pollution can affect not just the presence and abundance of specific insect taxa but also their ability to perform ecosystem services through altering their physical status [176] at levels ranging from the individual to the cellular. Fine sediments cause river impairment with consequences for freshwater insect abundance [177,178], functional traits [179], biomass [180], and species richness [181]. Even in cases where pollutants may not affect mayflies directly, individuals that have accumulated pollutants in their bodies may play a role in problems associated with others parts of ecosystems. For example, as part of their role in cycling matter between aquatic and terrestrial habitats, mayflies accumulate heavy metals from the water and sediment and transfer them elsewhere [182]. Heavy metals sometimes are lost during metamorphosis, however. It also is worthwhile to note that differences in tissue concentrations may exist between males and females [183]; such differences should be explored, along with any ramifications associated with differential dispersal of males and females.

Despite the introduction of legislation such as the Water Framework Directive in Europe and the Clean Water Act (CWA) in the USA, pollution continues to impact water quality in rivers around the world. Almost half of the sites monitored across Europe continue to suffer from chronic chemical pollution leading to long-term negative impacts on freshwater organisms [184]. One in ten sites suffered acute pollution with potentially lethal impacts for freshwater organisms. Sources of pollution included domestic and industrial sewage effluents and run-off from agriculture and urban areas, with pesticides posing the most acute risk to freshwater ecosystems. The impact of neonicotinoid pesticides is particularly worrying (e.g., [185,186]).

Diversity and overall abundance of freshwater invertebrates were significantly reduced in water chronically polluted with the pesticide imidacloprid [187]. Mayflies, caddisflies, and true-flies are particularly sensitive to these pesticides; even at low concentrations, there is a considerable risk of widespread impact on freshwater invertebrate populations [188]. In laboratory tests, half of mayflies and caddisflies died when exposed to concentrations in the range 0.1–0.3 μg/L; at just 0.03 µg/L 10% of mayflies died. Sub-lethal effects on invertebrates have also been detected, including changes in feeding rates, mobility, predation rates, reduced growth, and reduced emergence at levels between 0.3 and 1.5 μg/L [189]. Monitoring of watercourses in the UK has shown surprisingly high concentrations of imidacloprid in urban catchments, and it is suggested that domestic pet flea (Siphonaptera) treatments may be the mostly likely source [190].

A recently published study of waters in Australia [191] showed the prevalence and diversity of pharmaceuticals present in the stream and riparian food webs, especially among aquatic insects and spiders. More work is needed to study the movement of these compounds through connections of aquatic and terrestrial food webs; mayflies no doubt play a role here, linking pieces of the landscape mosaic. Beyond pharmaceutical pollution, general organic pollution may have complex effects; for example, further research is required to evaluate the importance of reduced zinc bioavailability associated with increased organic matter and water hardness to the species-poor communities in organic-contaminated rivers [192].

### 4.2. Invasive Alien Species

Invasive Alien Species (IAS) are an increasing threat throughout the World. In Europe, many of these non-native species originate from the Ponto-Caspian region, with over a hundred species known to have spread from this area to date [193]. The introduction of these non-native species to new ecosystems ultimately leads to a reduction in species richness and abundance, with mayflies, caddisflies, freshwater shrimps and other crustaceans particularly vulnerable. A list of 100 of the World’s worst invasive species features 9 freshwater invertebrates, including the Chinese mitten crab (*Eriocheir sinensis* H. Milne-Edwards (Decapoda: Varunidae)), the Fish-hook water flea (*Cercopagis pengoi* (Ostroumov) (Cladocera: Cercopagididae)) and Golden apple snail (*Pomacaea canaliculata* (Lamarck) (Gastropoda: Ampullariidae)) [194]. In Europe, crayfish species pose a particular threat, especially Signal crayfish (*Pacifastacus leniusculus* (Dana) (Decapoda: Astacidae)) and Red swamp crayfish (*Procambarus clarkii* (Girard) (Decapoda: Cambaridae)). The annual cost incurred due to damage caused by and/or the control of these species has been estimated at €454 million [195]. In the UK, a list of 56 invasive alien invertebrate species features 24 freshwater species [196], whereas, at the scale of Europe, more than 750 alien freshwater species have been recorded [197].

### 4.3. Habitat Loss and Degradation

Freshwater habitats are the most extensively and rapidly altered ecosystems on the planet, and they demonstrate broad response to these modifications, including changes to physical structure, chemistry, biotic characteristics and ecosystem processes [198].

Continued investment in hydro-power electricity generation is predicted to result in a 21% decrease in the number of remaining free-flowing rivers around the world. The majority of these developments are focused on the Amazon in South America, the Ganges in India, and the Yangtze in China [199]. Once impoundments are established, dam failures pose risks not only for humans downstream but also for aquatic life, including mayflies, both up- and downstream. This has been indicated for a recently discovered mayfly species, *Tricorythodes tragoedia* Souto, Angeli & Salles [200]. River drainage and flood-protection schemes involving the alteration of watercourses also cause changes to habitat conditions.

Dams and diversions may lead to changed conditions both up- and downstream, resulting in a changed habitat that may no longer support populations once prevalent [201]. Fragmented and isolated populations may be more prone to the effects of extreme genetic drift or extirpation when subjected to sporadic droughts and scouring from floods, which is especially problematic for species that inhabit marginal, shallow, and erosional zones of streams.

Development of urban areas and transportation infrastructure also impacts freshwater invertebrates in many ways. Perhaps seldom considered, the steady increase in the intensity and distribution of lights next to rivers may have a negative impact on populations. The adults of many species are attracted to light and bankside lights may lure them away from their natural waterside habitat (e.g., [202]). Similarly, asphalt roads can act as an ecological trap for mayflies, which are attracted to the horizontally polarized light reflected from their surfaces [203]. Solar panels are known to cause the same phenomenon [204], and the proliferation of this renewable energy source in recent years is a cause for concern, from the perspective of Ephemeroptera conservation. Fortunately, relatively simple mitigation measures can reduce the attractiveness of these panels [205].

A further impact of development is the placing of bridges over watercourses [206]. Upon approaching a bridge, up to 86% of Long-tailed mayflies (*Palingenia longicauda*) turn back rather than cross the bridge to continue upstream [207]. This is particularly problematic as it disrupts the compensatory upstream mating flight of the mayfly, thus restricting its range in the river. Further studies should explore the behaviors of other species with respect to bridges. Also, involving roadways in part, salinization of freshwaters is being seen as an emerging threat [208]. Baseline toxicity studies on pollution with fracking wastewater on three mayfly species in the Delaware river basin found chronic lethal effects after exposure over 20–30 days with a concentration of 0.5% produced water. In addition, non-lethal effects, including reduced reproductive rate and smaller adult size, were observed [209].

Degradation and loss of terrestrial habitat also impact mayfly communities. In addition to declines in water purification and flood mitigation, terrestrial degradation and loss via water erosion impact mayfly larvae. As silt and sediment loads increase in water bodies, larval populations of taxa intolerant to these loads (perhaps the majority of taxa) are challenged, but a few others, such as *Caenis* and some *Eurylophella* Tiensuu (Ephemerellidae) (genera with operculate gills) and many burrowers (which prefer soft substrate) benefit, either due to competitive release or more abundant potential habitat.

### 4.4. Climate Change

Climate change is widely recognized as being one of the major long-term threats to biodiversity [210]. Most recent predictions are that the average global temperature will continue to rise as a result of climate change, and this will inevitably have an impact on invertebrate populations. Indeed, with the majority of mayfly species having relatively short life cycles and good powers of mobility, they are likely to be one of the first groups to show the impact of a changing climate [211,212]. Cold-loving species may retreat northwards and uphill, while warm-loving species may increase their range [210]. However, many tropical and equatorial species evolved within narrow thermal regimes and already are living near their thermal maxima; even slight changes may outpace their ability to move or otherwise adapt [210]. An analysis of European Trichoptera species traits found that the biggest potential impact from climate change was likely in Southern Europe with up to 30% of the fauna in the Iberic-Macaronesian region being potentially endangered by climate change [213]. In the UK one study [214] found that a 3 °C rise in temperature could result in a 10–43% reduction in macroinvertebrate abundance in upland circumneutral streams and lead to the local extinction of at least the Gold-ringed dragonfly (*Cordulegaster boltonii* (Donovan) (Odonata: Cordulegastridae)), a caddisfly species (*Rhyacophila munda* R. McLachlan (Trichoptera: Rhyacophilidae)), and Pea mussels (species of *Pisidium* Pfeiffer (Bivalvia: Sphaeriida: Sphaeriidae)). The Upland summer mayfly (*Ameletus inopinatus* Eaton (Ameletidae), also known as the Holarctic comb minnow mayfly)—a predominately montane mayfly species restricted to coldwater streams—is now absent at many of its historical sites at lower altitudes, and some evidence suggests that it is being pushed further and further upstream as water temperatures rise [215]. European research using climate change models has shown that the geographical range of this species is likely to contract with remaining populations predicted to be restricted to the Alps, Scandinavia and parts of the Scottish Highlands such as the Cairngorms by 2080 [216]. In contrast, another study [217] found that numbers of emergent Ephemeroptera were unaffected by brownification and warming, under experimental conditions in a large-scale outdoor pond facility; this may suggest some differences between potential responses of lentic and lotic mayfly species assemblages that warrant further investigation.

It is unclear how an expanded season of meltwater will affect mayfly communities in streams fed in this way. More meltwater may lead to overall cooler conditions for these streams, rather than warmer conditions, for at least some of the year, and it may possibly lead also to periods of less thermal fluctuation than normal. All of these changes may pose challenges to the phenologies of species that have evolved in these streams, especially timing aspects related to egg diapause; alternatively, it might lead to range expansions or increases in population sizes.

## 5. Conclusions

Despite being a relatively small order of insects, mayflies deliver a wide variety of direct and indirect ecosystem services. They are excellent indicators of the condition of their habitats, in addition to delivering services such as providing food for humans and other animals, reworking sediments, decomposing wood and other vegetation, and purifying water through filter feeding. Future study of their feeding behaviors and general habits may reveal further ecosystems services provided by mayflies.

However, the continued provision of these services may be jeopardized by anthropogenic impacts such as pollution, invasive alien species, habitat loss and degradation, and climate change. Further efforts to assess the conservation status of mayfly species worldwide are required, and where appropriate, action should be taken to ensure that species are resilient to these impacts.

Concerted and coordinated actions worldwide to minimize, manage, and mitigate these impacts are necessary so that future generations may continue to benefit from the services provided by Ephemeroptera.

## Figures and Tables

**Figure 1 insects-10-00170-f001:**
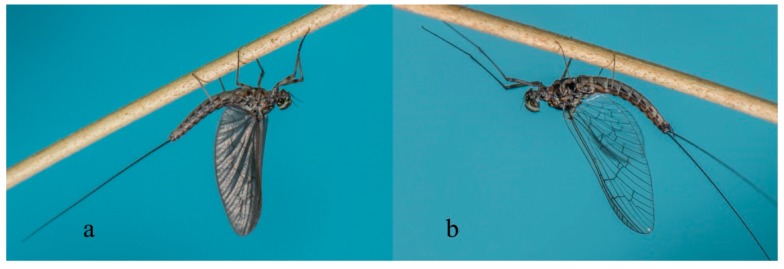
*Cloeodes penai* (Morihara & Edmunds) (Baetidae, Brazil) male subimago (**a**) and imago (**b**). Photograph courtesy of Frederico Salles.

**Figure 2 insects-10-00170-f002:**
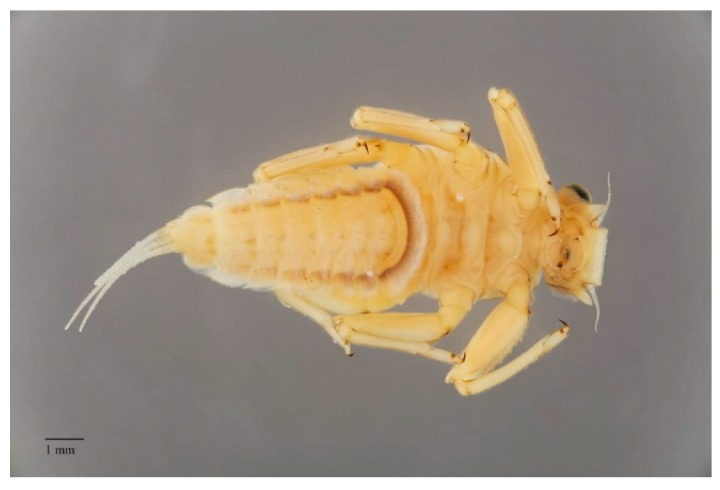
Dodds’ spiny crawler mayfly, *Drunella doddsii* (Needham) (Ephemerellidae, USA) in ventral view; abdominal sternites with rows of setae as a friction disk to adhere to the substrate in a fast flowing stream.

**Figure 3 insects-10-00170-f003:**
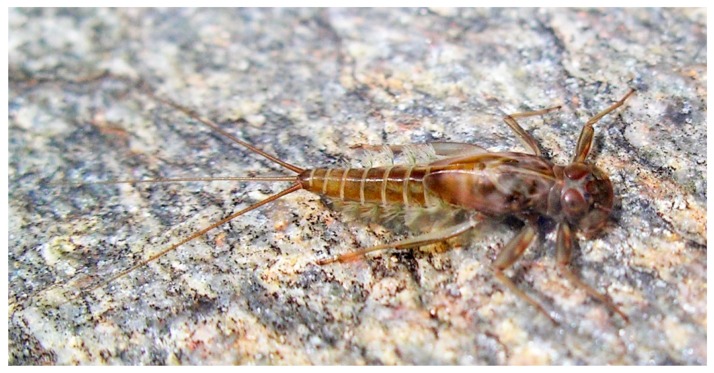
*Rhithrogena loyolaea* Navás (Heptageniidae, Switzerland) in dorsolateral view in its natural habitat; the flattened body, strong legs and first pair of gills functioning as a suction disk allow this species to withstand very strong current speed. Photograph courtesy of Laurent Vuataz.

**Figure 4 insects-10-00170-f004:**
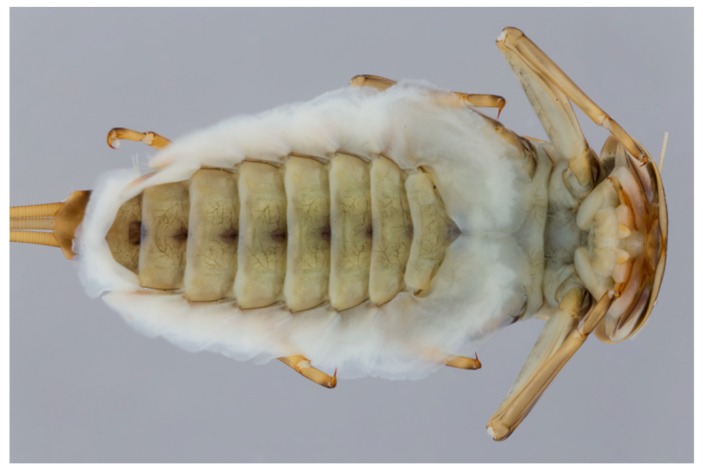
*Rhithrogena loyolaea* Navás (Heptageniidae, Switzerland) in ventral view to show the gills adapted to function as a suction disk.

**Figure 5 insects-10-00170-f005:**
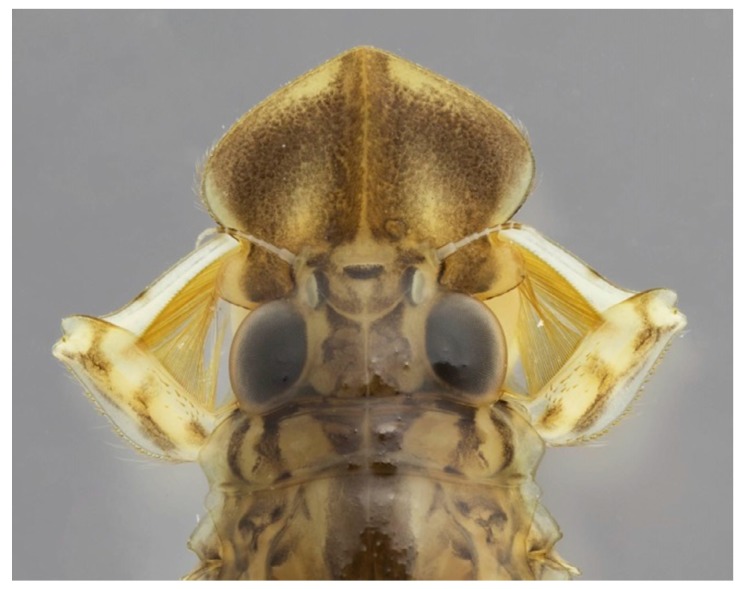
*Oligoneuria mitra* Salles, Soares, Massariol & Faria (Oligoneuriidae, Brazil) detail of head and prothorax in dorsal view; the first pair of legs equipped with filtering setae.

**Figure 6 insects-10-00170-f006:**
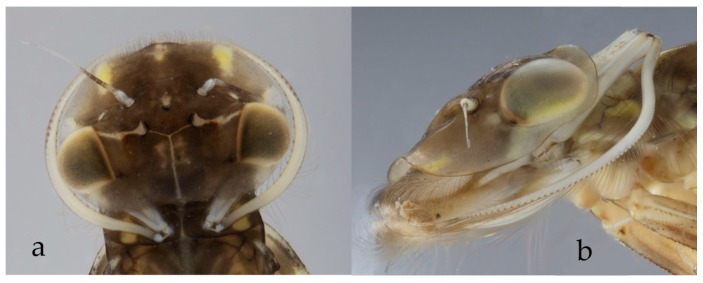
*Arthroplea congener* Bengtsson (Heptageniidae, Switzerland) in dorsal (**a**), and lateral (**b**) view: head, with maxillary palps clearly visible on the sides that are used to create filtering vortices.

**Figure 7 insects-10-00170-f007:**
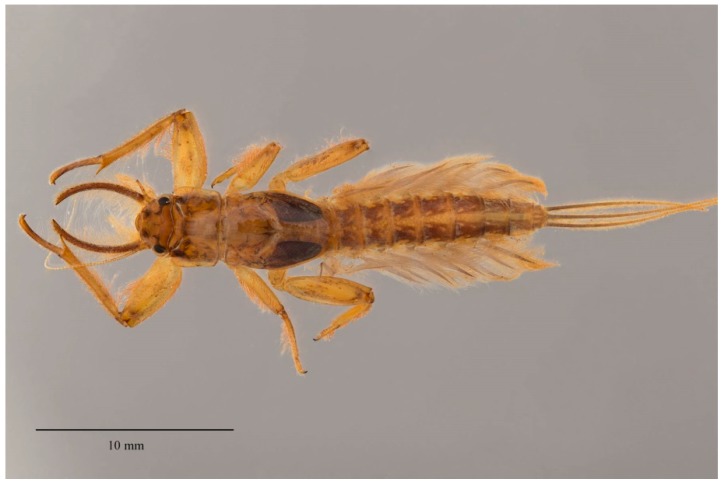
Ultimate larval instar of *Polyplocia campylociella* Ulmer (Euthyplociidae, Borneo) in dorsal view; a semi-burrower that uses its long mandibular tusks to dig soft substrate under cobble.

**Figure 8 insects-10-00170-f008:**
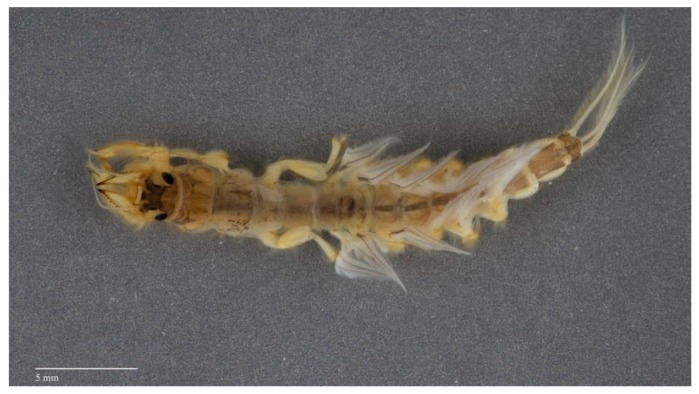
Larva of *Cheirogenesia edmundsi* Sartori & Elouard (Palingeniidae, Madagascar) in dorsal view; a burrower which digs into hard substrates, such as clay.

**Figure 9 insects-10-00170-f009:**
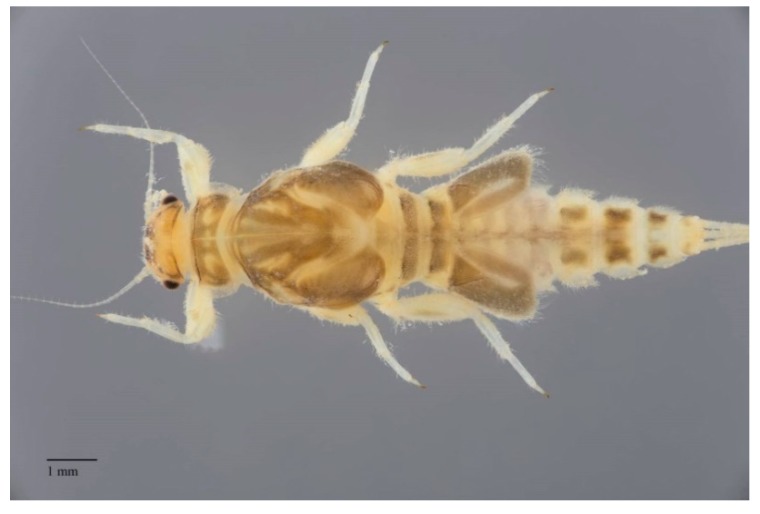
*Caenis luctuosa* (Burmeister) (Caenidae, Switzerland) in dorsal view; the second pair of gills is operculate and protects the delicate subjacent ones.

**Figure 10 insects-10-00170-f010:**
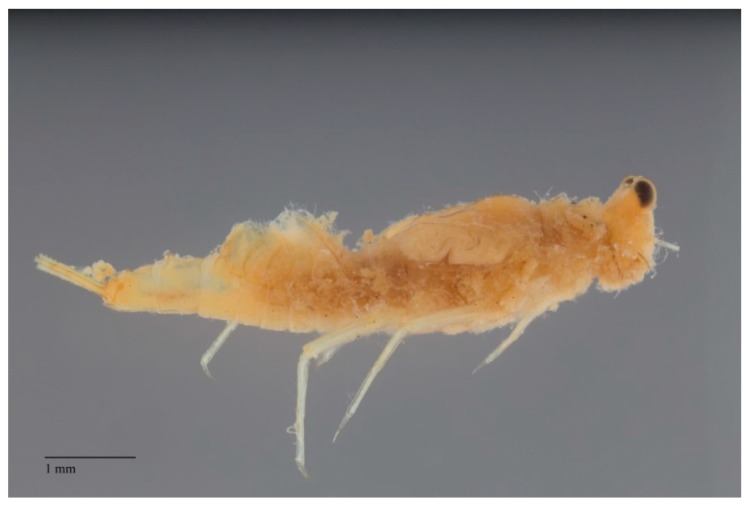
*Machadorythus maculatus* (Kimmins) (Machadorythidae, Ivory Coast) in lateral view; with middle abdominal segments modified to form a gills basket.

**Figure 11 insects-10-00170-f011:**
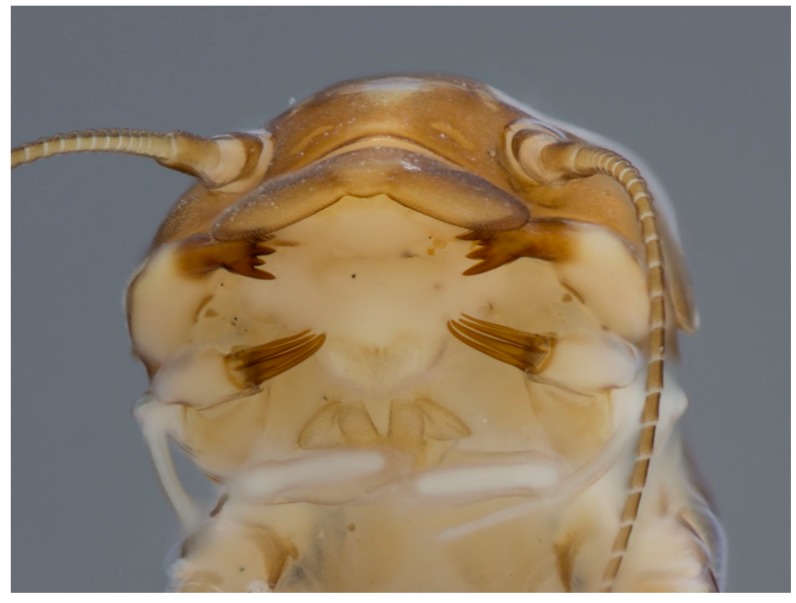
*Guloptiloides gargantua* Gattolliat & Sartori (Baetidae, Madagascar) details of the head in ventral view; mandibles and maxillae blade-like to prey mainly on other baetid larvae.

**Figure 12 insects-10-00170-f012:**
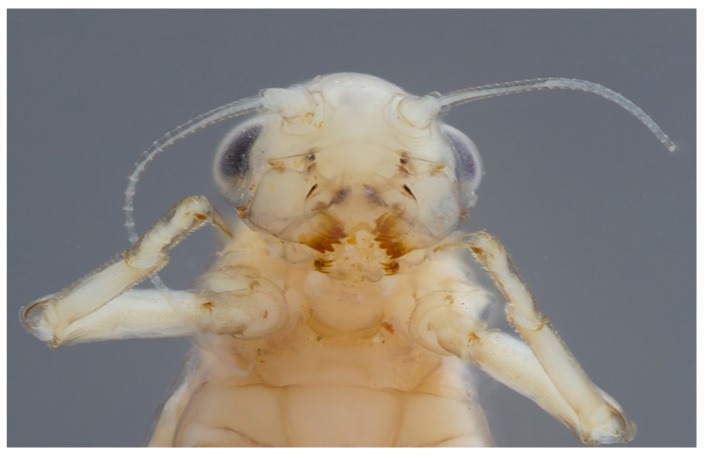
*Edmulmeatus grandis* Lugo-Ortiz & McCafferty (Baetidae, Madagascar) detail of head in ventral view; mandibles are cricket-like; this species shreds exclusively on *Hydrostachys* Thouars (Cornales: Hydrostachyaceae) aquatic plants and is completely green when alive.

**Figure 13 insects-10-00170-f013:**
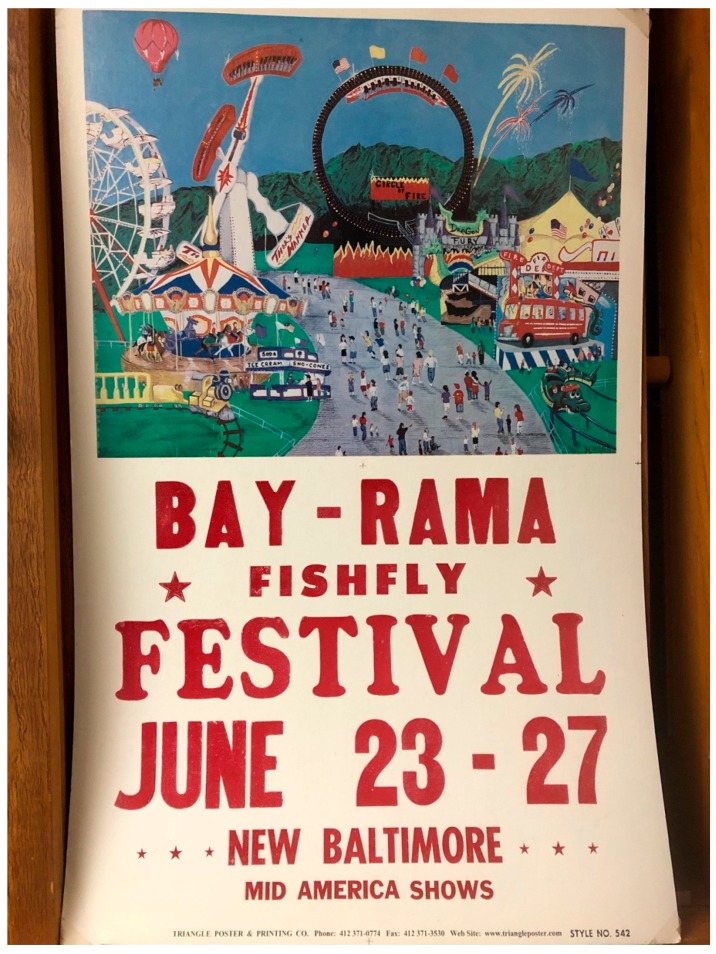
Advertising poster from “fishfly” festival in New Baltimore, Michigan, USA, ca. 2004. The annual celebration usually coincides with the mass emergence of Ephemeridae from Lake Saint Clair, which is located on the Michigan, USA and Ontario, Canada borderline.

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
