# Peer review of "Mayflies (Ephemeroptera) and Their Contributions to Ecosystem Services"

_insects, 2019, doi:10.3390/insects10060170_

Round 1

Reviewer 1 Report

Dear authors,

I congratulate you on the manuscript. Magnificent work. However, I listed below some minor remarks to be reviewed.

Line

Description

16

I feel the necessity to be clear about the aims of the paper, even though abstract indirectly says.

47

I am missing the authorship of each species, as well as the family's name of them.

111

I suggest the authors include a photo or even a drawn of the first pair of modified gill. Only by the figure, is difficult to understand how the shape is.

113

Coloburiscus spp., instead of Coloburiscus spp,

139

I recommend a photography or a drawn of the genus cited, especially to contrast with the example of the operculum the Caenidae and Ephemerellidae.

197

There is a reference for this information, is not there?

235

Just an observation. In Brazil, Alagoas state, there are also an annual festival for mayfly.

288

e.g. instead e.g

322

The expression “by work” could be suppressed, since it indicated the reference [79] only.

660

Is just this the reference?

718 and 850

The tittle of the paper is in uppercase letter.

898

If I am right, the correct is to cite de total number of the pages, not the cited page, as in the reference 184. If 130 is the total number of pages, please, switch the p. for pp. Review in the other references.

997

Number of pages is missing.

1026

There must be a printing error on the text.

Author Response

16 

I feel the necessity to be clear about the aims of the paper, even though abstract indirectly says. 

We  clarified the aims of the paper by adding a sentence to the beginning  of the abstract and providing more detail in the Introduction. 

47 

I am missing the authorship of each species, as well as the family's name of them. 

These have been added. 

111 

I  suggest the authors include a photo or even a drawn of the first pair  of modified gill. Only by the figure, is difficult to understand how the  shape is. 

We added a new figure. 

113 

Coloburiscus spp., instead of Coloburiscus spp, 

Corrected. 

139 

I recommend a photography or a drawn of the genus cited, especially to contrast with the example of the operculum the Caenidae and Ephemerellidae. 

We added a figure, in lateral view, of a species of Machadorythidae. 

197 

There is a reference for this information, is not there? 

We added the reference. 

235 

Just an observation. In Brazil, Alagoas state, there are also an annual festival for mayfly. 

We included this information. 

288 

e.g. instead e.g 

Corrected. 

322 

The expression “by work” could be suppressed, since it indicated the reference [79] only. 

Changed. 

660 

Is just this the reference? 

All references have been checked and corrected. This response applies to all remaining Reviewer 1 comments, too. 

Reviewer 2 Report

I was really excited about this paper when I first read the title and I hoped that it would be a worthy addition to the group of milestone review articles on mayflies that have stimulated many important life history and ecological studies over the past 35 years. Unfortunately, after reading through the manuscript I must declare that I find myself disappointed. Despite the fact that there are several good reviews of mayfly biology already in print the authors seemed to think another was necessary and essentially duplicated the content of these other works. To their credit they did include several excellent photographs, but those alone are not sufficient to claim a significantly different content. Moreover, this treatment seems a superfluous part of a review paper on ecosystem services. It is sufficient to presume that any reader of such a paper would take it upon themselves to gain the necessary background on mayflies to comprehend the content. In addition photos in figures 1 & 2 have been previously published in the Ephemeroptera chapter (page 878) of the 4th edition of Thorp and Covich's Freshwater Invertebrates: Ecology and General Biology Vol. 1 (2015). Thus because of existing reviews of mayfly biology this entire section should be deleted.

            Further into the manuscript the reader encounters a section on mayfly phylogeny, also seemly a superfluous part of a review paper on ecosystem services. Moreover there are acceptable treatments of the current phylogeny of Ephemeroptera already in print elsewhere. Again this entire section should be deleted. The paper should logically begin with what is identified as section 3 (Ecosystem Services). Perhaps it is logical to assume that anyone interested in ecosystem services would already be familiar with the 2005 Millennium Report on Ecosystem Services, but I recommend providing an introductory statement at the start of this section that introduces the concept and describes the categories to follow (which seem to be in accordance with the 2005 report.

            Starting from section 3 on the manuscript seems to have several instances of vague or imprecise statements. The most critical of these are marked by sticky notes on the pdf review copy. There are some instances of statements that seem to imply that there is a factual basis for some aspects of mayfly biology when in fact we have little or no data to support such claims. For example, the statements about mayfly larva (or nymphs if you like) purifying water by consuming either FPOM or CPOM. Even the once great densities of Hexagenia larva on the bottom of the Mississippi River near Winona Minnesota, USA were not enough to significantly change the load of organic material in those parts of the river system. Currently I know of no study that empirically shows a significant "water purification" effect exclusively from the organic feeding activities of mayflies. Other important effects of benthic burrowing species concerning the mobility of complexed nutrients (i.e., nitrogen and phosphorus compounds) are quickly stepped over. Much more could be said about these events. Also important nutrient transport and use events that are part of nutrient spiraling pathways are not elaborated on when they should be. There are allusions made about mayfly use  as a food for large carnivores as if the phenomena really hasn't been observed, when it has. There was an inappropriate use of the term "vector" when a more appropriate term was available. Along these lines there are far too many specific comments to list here. The  authors are referred to the comments on the manuscript.

            Perhaps the last problem I should mention here relates to the beginning of section 4 (Challenges to Mayfly Services). Here the authors make statements about losses of diversity across major biogeographic regions for insects in general and then without any mayfly specific data try to suggest these values should also apply to mayflies. This is quite a major leap in the dark without a net. There are well documented instances of aquatic habitat loss where mayflies where undoubtedly affected, but none of these events has shown effects at the continental scale as the authors imply. Data from Europe certainly is useful because of the long-term datasets that have been amassed, but criteria of judging imperilment need to be explicitly presented because they have changed over time. Only knowing such criteria can comparisons be made with other areas. In addition, this is such an important subject unpublished data can not be part of the conversation – publish it then everyone can evaluate it.

            In closing I think that a manuscript such as this has merit as a review of the current state of knowledge and an entry point into the primary literature for the specific topics mentioned. It could also play a role in facilitating future studies as other broad literature reviews on mayflies have done previously. However, I think much work is needed to bring the manuscript up to the level of something worthy for publication.   

Author Response

I was really excited  about this paper when I first read the title and I hoped that it would  be a worthy addition to the group of milestone review articles on  mayflies that have stimulated many important life history and ecological  studies over the past 35 years. Unfortunately, after reading through  the manuscript I must declare that I find myself disappointed. Despite  the fact that there are several good reviews of mayfly biology already  in print the authors seemed to think another was necessary and  essentially duplicated the content of these other works. To their credit  they did include several excellent photographs, but those alone are not  sufficient to claim a significantly different content. Moreover, this  treatment seems a superfluous part of a review paper on ecosystem  services. It is sufficient to presume that any reader of such a paper would take it upon themselves to gain the necessary background on mayflies to comprehend the content. In addition photos in figures 1 & 2 have been previously published in the Ephemeroptera chapter (page 878) of the 4th edition of Thorp and Covich's Freshwater Invertebrates:  Ecology and General Biology Vol. 1 (2015). Thus because of existing  reviews of mayfly biology this entire section should be deleted.  

We  replaced those figures. The general mayfly biology review section was  requested in the solicitation of this work by the guest editors, and so  we included it. In our revision of the manuscript based on this review  and four other peer-reviews, we attempted to tie together the various  sections of the paper better, in terms of the links between general  attributes of mayflies and how these attributes enable and constrain the  services that can be provided by mayflies. Given that this paper is one  of a series about aquatic insects in general, we felt the generalized  approach was justified and provided the potential for a wider audience  to understand the information provided without additional reading.  Sometimes an approach that reaches the widest audience is needed,  especially if we, as scientists, hope to reach out to policy makers. 

             Further into the manuscript the reader encounters a section on mayfly  phylogeny, also seemly a superfluous part of a review paper on ecosystem  services. Moreover there are acceptable treatments of the current phylogeny of Ephemeroptera already in print elsewhere. Again  this entire section should be deleted. The paper should logically begin  with what is identified as section 3 (Ecosystem Services). Perhaps it  is logical to assume that anyone interested in ecosystem services would  already be familiar with the 2005 Millennium Report on Ecosystem  Services, but I recommend providing an introductory statement at the  start of this section that introduces the concept and describes the  categories to follow (which seem to be in accordance with the 2005  report.  

We  added the reference, and we hope this helps to provide better context.  The categories of services are considered by some to be general  knowledge at this point, as they are included in textbooks for a variety  of biology courses, ranging from introductory courses for non-majors to  advanced courses on climate change biology. However, as mentioned  above, we are aiming for a broad audience, and this reference will help  those who have not worked with the concepts before.  A challenge of this  paper has been deciding whether it is to be written for mayfly  specialists who need to learn about ecosystem services or for ecosystem  service specialists who need to learn about mayflies, or for people who  perhaps don’t specialize in either area. We retain the section on mayfly  phylogeny and systematics because it was requested in the solicitation  of the article and for the same additional reasons that we retain the  section on general biology (see comments above). 

            Starting from section 3 on the manuscript seems to have several instances of vague or imprecise statements. The most critical of these are marked by sticky notes on the pdf review copy. There  are some instances of statements that seem to imply that there is a  factual basis for some aspects of mayfly biology when in fact we have  little or no data to support such claims.  For example, the statements about mayfly larva (or nymphs if you like)  purifying water by consuming either FPOM or CPOM. Even the once great  densities of Hexagenia larva on the bottom of the Mississippi River near Winona Minnesota, USA were not enough to significantly change  the load of organic material in those parts of the river system.  Currently I know of no study that empirically shows a significant "water  purification" effect exclusively from the organic feeding activities of  mayflies. Other important effects of benthic burrowing species  concerning the mobility of complexed nutrients (i.e., nitrogen and  phosphorus compounds) are quickly stepped over. Much more could be said about these events. Also  important nutrient transport and use events that are part of nutrient  spiraling pathways are not elaborated on when they should be. There are  allusions made about mayfly use  as  a food for large carnivores as if the phenomena really hasn't been  observed, when it has. There was an inappropriate use of the term  "vector" when a more appropriate term was available. Along these lines there are far too many specific comments to list here. The  authors are referred to the comments on the manuscript.  

We  clarified and corrected the language (including vector), and we added a  few more references to provide basis for our points. We responded to  sticky notes with changes in the text. We included a better reference  that refers to mayflies being consumed by large carnivores. In one  place, we removed direct mention of purification and instead emphasized a  more general processing and removal of substances; we mentioned that  their role may be small. Elsewhere, we emphasized that purification  roles have been assumed; perhaps drawing attention to this will lead to  further studies that will confirm or refute the notion. Many topics are  “quickly stepped over” in this article, as it is admittedly a  superficial overview where depth was sacrificed painfully for breadth.  It is by design not a comprehensive review of narrow topics (even when  those narrow topics are incredibly important). Our casual mention of  nutrient spiraling without explanation in the first draft was  inappropriate to the points we were trying to make, so we removed this  phrase from the revised version to allow more emphasis on energy flow  and nutrient transport in general, which is what was intended. An  excursion into the role of mayflies in nutrient spiraling was tempting,  but it would be a distraction from the main points we were attempting to  illustrate, because it involves so many abiotic and biotic factors  other than mayflies. 

             Perhaps the last problem I should mention here relates to the beginning  of section 4 (Challenges to Mayfly Services). Here the authors make  statements about losses of diversity across major biogeographic regions  for insects in general and then without any mayfly specific data try to  suggest these values should also apply to mayflies. This is quite a  major leap in the dark without a net. There are well documented  instances of aquatic habitat loss where mayflies where undoubtedly  affected, but none of these events has shown effects at the continental  scale as the authors imply. Data from Europe certainly is useful because  of the long-term datasets that have been amassed,  but criteria of judging imperilment need to be explicitly presented  because they have changed over time. Only knowing such criteria can  comparisons be made with other areas. In addition, this is such an important subject unpublished data can not be part of the conversation – publish it then everyone can evaluate it.  

We  slightly modified the organization of this section, provided links  between our logical steps, emphasized the lack of conservation status  studies globally, made remark about differing conservation criteria, and  removed references to unpublished or preliminary data. 

            In closing  I think that a manuscript such as this has merit as a review of the  current state of knowledge and an entry point into the primary  literature for the specific topics mentioned. It could  also play a role in facilitating future studies as other broad  literature reviews on mayflies have done previously. However, I think  much work is needed to bring the manuscript up to the level of something worthy for publication.     

We  appreciate that Reviewer 2 indeed saw merit in this manuscript, and we  hope that the changes we have made make it more worthy of publication.  Each of the authors would have preferred to have spent more time and  done a more thorough  and in-depth review, but each of us had to balance our many other  commitments along with making changes in a timely manner, and we hope  our responses are sufficient. In some cases, we followed the generally  favorable consensus of reviewers 1, 3-5, while also trying to address  concerns of reviewer 2.

Reviewer 3 Report

    The paper's objectives are worthy of the effort given.  Readers unfamiliar with the ecological roles of mayflies will get a good, first exposure.  The paper is less practical as a full account of all related research, as indicated by the authors.

    There are sections of the manuscript that need better organization and paragraph flow.  There is not a sense of purpose for each paragraph that is clearly presented in its opening sentence.  In addition, simplification of sentence structure throughout the manuscript would make the paper more readable.  The authors make frequent use of qualifiers such as "may be", "also may" and "are reported to have".  In cases where uncertainty remains, they are appropriate. In other cases, they add to sentence complexity and reduce readability.

Author Response

The  paper's objectives are worthy of the effort given.  Readers unfamiliar  with the ecological roles of mayflies will get a good, first exposure.   The paper is less practical as a full account of all related research,  as indicated by the authors. 

We further emphasized this last point in the text. 

    There  are sections of the manuscript that need better organization and  paragraph flow.  There is not a sense of purpose for each paragraph that  is clearly presented  in its opening sentence.  In addition, simplification of sentence  structure throughout the manuscript would make the paper more readable.   The authors make frequent use of qualifiers such as "may be", "also may" and "are reported to have".   In cases where uncertainty remains, they are appropriate. In other  cases, they add to sentence complexity and reduce readability. 

We  improved organization of material to have a smoother flow of thought.  We added introductory sentences or short paragraphs where appropriate,  sometimes with additional references for context. We simplified language  and removed redundancies and superfluous phrases. Comments in the text  have been addressed in our revised manuscript. 

Reviewer 4 Report

The review article on mayflies and their importance for ecosystem is overall well written. The authors cover a lot of aspects on mayflies including its biology and life cycle, different cultural meanings, contributions of mayflies to the ecosystem and the challenges mayflies are facing. While it is important to do field work studies to understand more about environmental impacts on mayflies, In the last decade, lots of lab work has been done (i.e. in the Buchwalter lab) to study the physiology of mayflies under different pollution (i.e. metals) or thermal challenges. In order to understand better of mayflies, studies from the individual level or even as small at the cellular/molecular level are needed and should be acknowledged more in this review article. 

Author Response

The review article on mayflies and their importance for ecosystem is overall well written. The authors cover a lot of  aspects on mayflies including its biology and life cycle, different  cultural meanings, contributions of mayflies to the ecosystem and the  challenges mayflies are facing. While it is important to do field work  studies to understand more about environmental impacts on mayflies, In  the last decade, lots of lab work has been done (i.e. in the Buchwalter  lab) to study the physiology of mayflies under different pollution  (i.e. metals) or thermal challenges. In order to understand better of  mayflies, studies from the individual level or even as small at the  cellular/molecular level are needed and should be acknowledged more in  this review article.  

We now include such references, especially from the Buchwalter lab (e.g., Xie et al. 2010, Conley et al. 2013,  Chou et al. 2017, Sweeney et al. 2018) and explicit language referring  to the fact that pollution affects not only species and populations, but  also acts at levels ranging from the individual to the cell.

Reviewer 5 Report

It was a pleasure to read this paper, which provides a comprehensive review of the ecology, systematics, and morphology of mayflies. These kinds of reviews are difficult to write, but the authors did an excellent job in balancing the breadth of information into a concise and readable summary. While each of the areas of focus are not necessarily new (it's a review after all), I am unaware of a single source that adequately summarizes the information available in one place. This will be required reading for my students who study aquatic insects in the future.

I have only minor suggestions.

Figures 1-2. It was difficult to discern the subimago/imago stages. I think it would help to add the nymph stage and wingpad stage to this, so that each stage is from left to right. Also add the name of the stage directly to the photo, rather than just a 1 or 2.

L218: This section was particularly good. Many treatments of cultural services for animals seem generic. This one makes an impactful case.

L409:412: Heavy metals in particular are often lost during metamorphosis, which affects their ability to "transfer metals elsewhere". The larva to subimago stage is a transition in which metals are mostly lost, with some extra loss from subimago to imago. See Kraus, J. M., Walters, D. M., Wesner, J. S., Stricker, C. A., Schmidt, T. S., & Zuellig, R. E. (2014). Metamorphosis alters contaminants and chemical tracers in insects: implications for food webs. Environmental science & technology48(18), 10957-10965. Also the Wesner (currently cited as ref. 173) paper discusses this loss as well. 

Citations: Some of the citations are mis-formatted (e.g. title in call caps). This is a common bug in reference software. Be sure to fix before final submission.

Author Response

It  was a pleasure to read this paper, which provides a comprehensive  review of the ecology, systematics, and morphology of mayflies. These  kinds of reviews are difficult to write, but the authors did an  excellent job in balancing the breadth of information into a concise and  readable summary. While each of the areas of focus are not necessarily  new (it's  a review after all), I am unaware of a single source that adequately  summarizes the information available in one place. This will be required  reading for my students who study aquatic insects in the future. 

I have only minor suggestions. 

Thank you for this kind evaluation, and we are glad you find value in the work at hand. 

Figures 1-2. It was difficult to discern the subimago/imago stages. I think it would help to add the nymph stage and wingpad stage to this, so that each stage is from left to right. Also add the name of the stage directly to the photo, rather than just a 1 or 2. 

We  substituted new, clearer images of subimago and imago. We avoided words  superimposed on images, preferring clear and carefully composed  captions. We provide visual examples of various forms of larvae in whole  or part throughout. 

L218:  This section was particularly good. Many treatments of cultural  services for animals seem generic. This one makes an impactful case. 

Again, thank you for the kind words. We minimized changes to this section. 

L409:412:  Heavy metals in particular are often lost during metamorphosis, which  affects their ability to "transfer metals elsewhere". The larva to subimago stage is a transition in which metals are mostly lost, with some extra loss from subimago to imago. See Kraus, J. M., Walters, D. M., Wesner, J. S., Stricker, C. A., Schmidt, T. S., & Zuellig, R. E. (2014). Metamorphosis alters contaminants and chemical tracers in insects: implications for food webs. Environmental science & technology, 48(18), 10957-10965. Also the Wesner (currently cited as ref. 173) paper discusses this loss as well.  

We simply added another citation to the more recent Wesner et al. (2017) paper. 

Citations: Some of the citations are mis-formatted (e.g. title in call caps). This is a common bug in reference software. Be sure to fix before final submission. 

We addressed these issues. 

Round 2

Reviewer 2 Report

Second Review – Comments for authors:

            It is great to see that the authors have taken up the challenge of revising their manuscript and even though there are still a few areas that I think could still be improved, I am satisfied that the majority of my initial concerns has been addressed. For my part as a reviewer it would have been helpful to know initially that authors had included the initial review of mayfly biology at the request of the guest editors. Had I known that I probably would not have recommended it be removed. Although I think I now understand the goal of this paper a little better, I think the world has enough such reviews for now. As for the systematics section, the purpose of that was also not made clear in the information I had starting the review, hence it too seemed redundant. However, because it now seems that a primary goal of this paper is to open up the context of mayfly literature to a wider audience (perhaps technical or even a political) I do not object to its inclusion.

            Additionally the authors have addressed my concerns about comments regarding mayflies processing organic matter or water pollution remediation. Overall the authors have done an admirable job in describing what are generally known as the positive ecosystem services of mayflies and mayfly communities; however, there are a few negatives, but perhaps that is left to another time. To conclude I am satisfied with the corrections that have been made and recommend the manuscript for publication as per the marked changes in the review copy sent to me for this second review.

Author Response

Thank you for reviewing our manuscript a second time. We are glad that our changes addressed your concerns. Thank you for recommending it for publication.